# Exploring the Role of Guanylate-Binding Protein-2 in Activated Microglia-Mediated Neuroinflammation and Neuronal Damage

**DOI:** 10.3390/biomedicines12051130

**Published:** 2024-05-20

**Authors:** Ji-Eun You, Eun-Ji Kim, Ho Won Kim, Jong-Seok Kim, Kyunggon Kim, Pyung-Hwan Kim

**Affiliations:** 1Department of Biomedical Laboratory Science, Konyang University, Daejeon 35365, Republic of Korea; jean9643@naver.com (J.-E.Y.); kimej1910@gmail.com (E.-J.K.); 2Myunggok Medical Research Institute, College of Medical School, Konyang University, Daejeon 35365, Republic of Korea; kimong104@naver.com (H.W.K.); jskim7488@konyang.ac.kr (J.-S.K.); 3Department of Digital Medicine, College of Medicine, University of Ulsan, Seoul 05505, Republic of Korea; kkkon1@amc.seoul.kr; 4Department of Convergence Medicine, Asan Medical Center, Seoul 05505, Republic of Korea

**Keywords:** neuron damage, guanylate-binding protein 2, immune response, activated microglia, neurodegenerative disease

## Abstract

Neuron damage by microglia, which act as macrophage cells in the brain, can result in various brain diseases. However, the function of pro-inflammatory or anti-inflammatory microglia in the neurons remains controversial. Guanylate-binding protein-2 (GBP2) is expressed and activated in the microglia in the early phase of the inflammatory response and plays an important role in controlling immune responses. In this study, we evaluated whether GBP2 initially reduces the immune response induced by microglia, and whether microglia induce pro-inflammatory functions in neurons via GBP2 expression. In lipopolysaccharide (LPS)-stimulated microglia, we assessed the expression of GBP2 and how it affects neurons via activated microglia. The biological functions of microglia due to the downregulation of the *GBP2* gene were examined using short hairpin RNA (shRNA)-RNA-GBP2. Downregulated GBP2 affected the function of mitochondria in the microglia and showed reduced neuronal damage when compared to the control group in the co-culture system. Furthermore, this protein was observed to be highly expressed in the brains of dementia mice. Our results are the first to report that the downregulation of GBP2 in activated microglia has an anti-inflammatory function. This study suggests that the *GBP2* gene can be used as a therapeutic target biomarker for inflammation-related neurodegenerative diseases.

## 1. Introduction

Neuroinflammation, known as inflammation of the central nervous system (CNS), is a prevalent and persistent clinical problem, due to our incomplete understanding of its pathogenesis [1,2]. Evidence strongly suggests that neuroinflammation is a pivotal process involved in the progression of neuronal degeneration, a common feature observed in several neurodegenerative disorders. Previously, research on CNS diseases has been focused on neurons. In the last decade [3], research has indicated that microglia, as the representative immune cells in the CNS [4,5], play important roles in neuropathic disease. However, the specific molecular and cellular relationships between microglia and neurons are still unclear [6]. Under steady-state conditions, microglia are key regulators of neural homeostasis owing to their more neurotrophic and less pro-inflammatory properties [7,8] in the brain (neuroprotective) [9,10] and are involved in various CNS diseases, such as pain [11], infection [12], neurodegeneration [13], brain trauma [14], and stroke (neurotoxic role) [15]. Nevertheless, during pathological conditions induced by trauma, infection, or the infiltration of autoimmune cells, this balance is disturbed, leading to the overactivation of microglia, and alterations to their phenotype and function, which can be either beneficial or destructive [14,16,17]. Although proper activation of microglia can be beneficial for microenvironment reconstruction, if their pro-inflammatory responses are excessive, they can aggravate the damage [18]. 

The role of microglia in pro- or anti-inflammation remains controversial. Importantly, depending on the integration of regulatory signals, microglial cells may undergo two different kinds of activation, acquiring a neurotoxic phenotype or a neuroprotective phenotype, and these have been called “classical” activation (M1) and “alternative” activation (M2) phenotypes, respectively, which are analogies for the phenotypes in peripheral macrophages [18,19]. While M1-like microglia generate a detrimental microenvironment for neurons by producing inflammatory cytokines and reactive oxygen species (ROS) when microglia are exposed to foreign materials [19], such as lipopolysaccharide (LPS) or interferon-gamma (IFN-γ), the M2 phenotype is involved in brain repair and regeneration by enhancing phagocytosis [20,21], releasing neurotrophic factors, and reducing brain inflammation. Activated microglia can also be identified by using the concomitant upregulation or de novo synthesis of a variety of cell surface and cytoplasmic molecules. 

The ability to readily identify activated microglia has provoked considerable interest in their value, as indicators of pathology with possible diagnostic potential [19]. If we could characterize a unique gene expression signature of polarized microglia, it would be notable for both microglia biology and microglia-targeted therapies for various CNS diseases. Therefore, we analyzed a variety of the substances that were secreted from microglia that had been stimulated by foreign pathogens to perform their own functions according to the circumstances. Among them, we focused on the role of guanylate binding protein 2 (GBP2) as a candidate genetic material involved in the regulation of polarization, metabolic reprogramming, and inflammatory reactions that cause microglia to damage nerve cells [22,23]. Guanylate binding proteins (GBPs) are members of the 65–67 kDa GTPase family, and those induced by IFNs confer protectant functions against phylogenetically diverse pathogens and cooperate with core production and pyroptosis [24,25]. 

There are reports that GBP2 is highly expressed in representative brain disorders such as Alzheimer’s disease [26]. This gene was identified as one of the most highly expressed genes in genomic analysis of brain tissue, indicating that its high expression reflects induced immune responses. Recent reports have shown that GBP2 exhibits high expression in M1 microglia, one of the two phenotypic states of microglia [27]. This demonstrates that *GBP2* can act as a gene-inducing immune response in microglia, the immune cells of the brain. However, it remained unclear whether the regulation of GBP2 expression impacts the function of microglia. 

Therefore, we aimed to investigate the role of the *GBP2* gene in neurodegenerative diseases. Specifically, we examined the phenotypic changes in microglia resulting from the downregulation of GBP2 expressed in M1 microglia and to demonstrate whether microglia activated by external pathogens possess pro-inflammatory functions that contribute to neuronal damage, and to evaluate the impact of GBP2 downregulation in these activated microglia on neuronal cells (Figure 1).

## 2. Materials and Methods

### 2.1. Cell Cultures

BV2 microglial cells and SH-SY5Y neuroblastoma cells were kindly provided by Professor Yang Seung-Joo of Konyang University. These cell lines were cultured in high-glucose Dulbecco’s modified Eagle’s medium (DMEM; Hyclone, Logan, UT, USA) supplemented with 10% FBS (Hyclone) and 1% penicillin G-streptomycin solution (Gibco, Grand Island, NY, USA) at 37 °C and 5% CO_2_.

### 2.2. RNA Extraction and Relative Quantitative Polymerase Chain Reaction (PCR) and Conventional PCR

BV2 microglial cells were seeded in three independent cultures and treated for 4 h with 1000 ng/mL of LPS. Total RNA was isolated using TRIzol reagent (Invitrogen, Waltham, MA, USA), and reverse transcription was performed using a complementary DNA (cDNA) synthesis kit (Solgent, Daejeon, Republic of Korea). Conventional PCR products were electrophoresed on 2% agarose gels prepared in 1 × TAE buffer (Biosesang, Yongin, Republic of Korea). Gel images were obtained using a chemiluminescence analyzer (VilberLourmat, Eberhardzell, Germany). Quantitative real-time PCR was performed using the Solg^TM^ 2× RT-PCR Smart Mix (Solgent). The comparative Ct (2^−ΔΔCt^) method was used to analyze the relative expression of the mRNAs normalized to that of the *GAPDH*. Primer design and PCR amplification were performed as described in Table 1. All kits used in this study followed the manufacturer’s guidelines.

### 2.3. Transfection of shRNAs-GBP

Small hairpin RNA (shRNA)-GBP2 was constructed using the same sequence (NM_010260.1). To establish the conditions for GBP2 knockdown (Kd) and scrambles (Scr), shRNAs targeting GBP2 were subcloned into pGFP-C-shLenti shRNA Vectors (Origene, MD, USA) (Appendix A). BV2 microglial cells were seeded in 6-well plates in DMEM (Hyclone) containing 10% FBS (Hyclone) at 3 × 10^5^ cells/well for 24 h in 5% CO_2_ at 37 °C before transfection. Then, the cells were transfected with shRNAs-GBP2 using Mirus (Takara Mirus Bio, Madison, WI, USA) for 4 h in serum-free medium according to the manufacturer’s instructions. The media were changed 24 h post-transfection. To confirm the efficiency of GBP2 Kd, the harvested cells were validated by conventional PCR after RNA extraction.

### 2.4. Enzyme-Linked Immunosorbent Assay (ELISA) for IL-6, TNF-α, and IL-10

To examine the immune reaction levels of the activated microglia, BV2 microglial cells were seeded at a concentration of 3 × 10^5^ cells in a 6-well plate. The supernatants of BV2, BV2 activated by LPS, and activated BV2 cells treated with shRNA culture medium were collected. The expression levels of IL-10, IL-6, and TNF-α induced by BV2 microglia were determined using ELISA kits (R&D Systems, Minneapolis, MN, USA). The assay was carried out according to the manufacturer’s instructions and measured at 450 nm using a Versamax Microplate Reader (Molecular Devices, Sunnyvale, CA, USA). The results were calculated using standard values obtained from a linear regression equation.

### 2.5. MTT Assay

Cell viability was measured using a 3-(4,5-dimethylthiazol-2-yl)-2,5-diphenyl tetrazolium bromide (MTT; Sigma, Poole, UK) reduction assay. In brief, BV2 microglial cells or SH-SY5Y neuroblastoma cells were seeded in 24 well plates with cell densities of 60–70% per well for 24 h and then treated with LPS. The media were removed, and then 300 μL of MTT (Sigma, 2.5 mg/mL) solution was added to each well and the cells were left to incubate and respond at 37 °C for 4 h and then treated with 300 μL of dimethyl sulfoxide at 37 °C for 15 min, and the absorbance was measured at 570 nm using a VerxaMax microplate reader.

### 2.6. In-Cell Western (ICW) Assay for the Detection of Nuclear Factor-Kappa B (NF-κB)

To assess whether the various cytokines secreted from the activated microglia are caused by NF-κB activation due to GBP2 expression, an ICW assay was conducted using the Odyssey Imaging System (LI-COR Biosciences, Lincoln, NE, USA) according to the manufacturer’s instructions. Briefly, BV2 cells were seeded at 1 × 10^4^ cells/mL for 24 h in 96-well plates (Falcon™, Cat# 161093, BD Biosciences, San Jose, CA, USA; Nunc™, Cat# 161093, Thermo Fisher Scientific, Waltham, MA, USA), fixed with 3.7% formaldehyde for 20 min at room temperature (RT), and then blocked with LI-COR Odyssey Blocking Solution (LI-COR Biosciences) for 90 min. The cells were incubated at RT overnight with NF-κB pre-mixed with a mouse IgG antibody against β-actin (1:500 dilution, Santa Cruz Biotechnology, Dallas, TX, USA). After five washes with 0.1% PBST, the cells were stained with a goat anti-mouse IgG IRDye™ 680 antibody (1:800 dilution, LI-COR Biosciences) at RT for 1 h. The microplates were scanned using the Odyssey CLx Infrared Imaging System (LI-COR Biosciences), and the integrated fluorescence intensities representing the protein expression levels were acquired using the software provided with the imager station (Odyssey Software Version 3.0, LI-COR Biosciences). The relative amount of protein was obtained by normalizing to endogenous β-actin in all experiments.

### 2.7. Nitrate Assay

To evaluate the anti-inflammatory induction levels of the activated microglia, the accumulation of nitrite (NO_2_^−^) as an indicator of NO production was evaluated using the Griess reaction in culture supernatant fluids. BV2 microglial cells were seeded at 2 × 10^4^ cells/well in 96-well plates and transfected with shRNA-Scr or -GBP2 for 4 h. Fifty microliters of culture supernatant was mixed with 50 µL of Griess reagent at room temperature for 15 min, and the absorbance was measured at 540 nm using a VerxaMax microplate reader.

### 2.8. Extracellular Acidification Rate (ECAR) Test for Glucose Metabolism of Mitochondria in Neuronal Cells

SH-SY5Y neuroblastoma cells were assayed as described in the manufacturer’s instructions in XF DMEM (containing 2 mM glutamine, Agilent Seahorse, Agilent Technologies, Santa Clara, CA, USA) supplemented with 10 mM pyruvate and 25 mM glucose. SH-SY5Y neuroblastoma cells were plated at 2 × 10^4^ cells per well in 8-well Seahorse culture plates (Agilent Seahorse) in 200 µL of the appropriate growth medium. One day after plating, SH-SY5Y neuroblastoma cells were treated with conditioned media (BV2 activated by LPS and transfected with shRNA). We then measured ECAR on a Seahorse FXp, as described by the manufacturer, with modifications. The concentrations of the inhibitors used in this study were as follows: 20 mM glucose followed by 1 mM or 2 mM rotenone, and 50 mM or 100 mM 2-deoxyglucose.

### 2.9. Wright–Giemsa Staining

BV2 microglial cells were seeded at 4 × 10^4^ cells/well in 24-well plates for 24 h and then transfected with shRNAs and treated with LPS. The treated cells were incubated for another 24 h and then the cells were fixed with cold methanol and washed with phosphate-buffered saline (PBS). Giemsa staining (Xiangya, Wuzhou, China) solution was used to stain live or apoptotic cells and was photographed using an optical inverted microscope (Zeiss, Oberkochen, Germany).

### 2.10. Flow Cytometry for Apoptosis Analysis 

The cell death population of the neuronal cells induced by activated microglia treated with shRNA-GBP2 was quantitatively measured by double staining flow cytometry. Apoptosis was detected using the FITC Annexin V Apoptosis Detection Kit (BD, Franklin Lakes, NJ, USA), according to the manufacturer’s protocol. BV2 microglial cells and SH-SY5Y neuroblastoma cells were cultured in a co-culture system with microglia activated by LPS in the upper insert and SH-SY5Y neuron cells on the bottom plate. Neuronal cells stimulated by the activated microglia were washed twice with cold PBS, centrifuged, and resuspended in 1× binding buffer at a concentration of 1 × 10^6^ cells/mL, and then placed at RT for 15 min after adding 10 μL annecin V-FITC to the cell suspension. The cells were transferred to 100 μL of the solution (1 × 10^5^ cells/mL), and then 5 µL of annexin V-FITC and 5 µL propidium iodide (PI) were added and they were kept in the dark for 15 m. Next, 400 μL of binding buffer was added to the suspension. Apoptotic cells (annexin V-positive and PI-negative cells) were measured as the percentage of gated cells. The stained cells were analyzed using flow cytometry (NOVOVYTE flow cytometer, ACEA Bioscience Inc., Santa Clara, CA, USA).

### 2.11. Genomic DNA Extraction in Mice Brain Tissue with Alzheimer’s Disease

Mice with Alzheimer’s disease (5× FAD) and normal mice (WT) were kindly provided by Dr. Min-Ho Moon at Konyang University in Daejeon, Republic of Korea. After separating the frontal cortex (FC), hippocampus formation (HP), and olfactory bulk (OB) from the mice, genome DNA was extracted using the manufacturer’s recommended experimental method using the Exene Genomic DNA microkit (Geneall, Seoul, Republic of Korea) at each site. The mice used included a regular mouse (WT) and a mouse with Alzheimer’s disease (5× FAD mouse in Week 11). Then, conventional PCR (Solgent) was performed in accordance with the conditions of 95 °C for 3 min, 95 °C for 30 s, 72 °C for 30 s, 72 °C for 5 min, and 4 °C (∞), and 2% agarose were made, and then the target gene of GBP2 was identified. 

### 2.12. Statistical Analysis

All data are presented as mean SEM and were analyzed statistically using the SPSS software program (SPSS V20.2). All experiments were performed in triplicate at least. The experimental results were determined using a paired *t*-test. The significance of the analysis is shown in the figures as follows: * *p* < 0.05, ** *p* < 0.02, and *** *p* < 0.01. 

## 3. Results

### 3.1. Confirmation of Substantial GBP2 Expression in the Brain Tissue of Mice with Alzheimer’s Disease (AD)

To investigate whether GBP2 is highly expressed in an actual neurodegenerative disease like Alzheimer’s disease, we examined the expression of GBP2 in an Alzheimer’s disease mouse model. Three brain regions known to have a high presence of microglia, namely the frontal cortex (FC), hippocampus (HP), and olfactory bulb (OB), were extracted from both wild-type (WT) mice of the same age and mice with AD (5× FAD). Genomic DNA was extracted, followed by PCR. As shown in Figure 1A, it was observed that GBP2 expression was higher in the brain tissues of mice with AD compared to normal mice in the tested region. This observation was consistently quantified in the results (Figure 1B). Thus, it was verified that GBP2 expression was significantly elevated in the brains of mice with AD, suggesting its potential as a specific biomarker for degenerative brain disorders, as supported by our findings in this study.

### 3.2. Optimal Conditions for the Induction of GBP2 Expression in BV2 Microglia Cells

Based on the results of GBP2 expression in brain disease animal models (Figure 1), we evaluated whether this gene is expressed in microglial cells present in the brain. To optimally induce GBP2 expression in the microglia stimulated by a foreign pathogen, BV2 microglial cells were treated with LPS using various concentrations (10, 100, and 1000 ng/mL) and durations of exposure (from 1 to 5 h). GBP2 expression was significantly increased in the cells treated with 1000 ng/mL LPS when compared to the other concentrations (Figure 2A,B). In addition, when the cells were treated with an optimal concentration of LPS for 4 h, higher GBP2 expression was observed compared to that in the other groups (Figure 2C,D). All quantitative data for the *GBP2* band showed a similar tendency. These results indicate that LPS induces the expression of GBP2, leading to the induction of GBP2 expression.

### 3.3. Neuronal Cytotoxicity after Switching on M1-Type Polarization for Microglia via LPS Stimulation

Lipopolysaccharide (LPS) is a classic stimulus for M1-type conversions of microglia because it causes the M1 phenotype to express pro-inflammatory cytokines from its microglia. To confirm whether microglia activated by LPS are polarized into the M1 type, we examined the expression of the representative pro-inflammatory biomarkers, such as intercellular adhesion molecule-1 (ICAM1) and lipocalin 2 (LCN2). As shown in Figure 3, the expression levels of ICAM1 and LCN2 were significantly higher in the microglia treated with LPS than in the untreated microglia. The quantitative data for the ICAM1 and LCN2 bands also showed a similar tendency (Figure 3B,C). These results indicate that LPS stimulation converted microglia into the M1 type, suggesting that LPS triggered an inflammatory reaction in BV2 microglial cells.

To explore neuronal cell death caused by the activated microglia that had been converted to the M1 type, we used a co-culture system. There were BV2 microglial cells seeded in the upper well and SH-SY5Y neuroblastoma cells in the bottom. After co-culturing with the activated microglia for 1 d, neuronal cell death was assessed using the MTT assay. Both neurons and neurons co-cultured with inactivated microglia lacked severe neuronal toxicity, demonstrating that the inactivated microglia did not affect the neuronal cells (Figure 3D). However, severe cell damage was observed in the group that was co-cultured with activated microglia. Taken together, these results indicate that microglia activated by LPS were converted to the M1 type and then caused neuronal cell toxicity.

### 3.4. Change from M1 to M2-Type Polarization with GBP2 Knock-Down

Next, we evaluated how the GBP2 that was expressed by the activated microglia affected neuroinflammation and neuronal damage. First, we generated shRNAs to evaluate the biological functions of the GBP2 in microglia activated by LPS. To select shRNAs with optimal knockdown (Kd) efficiency, shRNA series were transfected into activated BV2 microglial cells. Among the tested shRNAs, shD-GBP2 showed a markedly higher Kd efficiency than the other groups (Figure 4A). Consistent with the results shown in Figure 4A, a similar tendency was observed (Figure 4B). Based on this result, we used shD as the best shRNA against GBP2 in all subsequent experiments. Next, we evaluated the characteristics of activated microglia according to GBP2 Kd. Our results revealed that shD-GBP2 exhibited lower expression of the pro-inflammatory markers (ICAM1 and LCN2) than shScr- or LPS-treated microglial cells (Figure 4C,D). Conversely, the expression of anti-inflammatory markers (ARG1 and IL-10) was increased when compared to the other groups (Figure 4E,F). 

Taken together, these results imply that the expression of GBP2 in LPS-activated BV2 microglial cells plays a role in pro-inflammatory function as it undergoes M1 polarization, which may affect neuronal viability. Furthermore, the downregulation of the *GBP2* gene induced the conversion from M1 to M2 polarization, indicating that the activated microglia will subsequently become deactivated cells.

### 3.5. The Induction of Anti-Inflammatory Function by the Down-Regulation of NF-κB via GBP2 Suppression

The expression of various pro-inflammatory cytokines is known to depend on the activity of NF-kB [28]. The Kd of *GBP2* genes resulted in a decrease in the immunological activity of microglia stimulated by LPS, leading to anti-inflammatory effects. Therefore, we evaluated how the downregulation of GBP2 affects the activity of NF-κB. To address this, the activation level of NF-κB was assessed using the ICW assay. As shown in Figure 5A, the upregulation of NF-κB in the BV2 microglial cells was observed in the cells treated with LPS or shScr compared with that of BV2 only (NC group). Interestingly, microglial cells treated with shD-GBP2 had decreased downregulation of NF-κB. The quantitative data of NF-κB in the Odyssey image showed a similar tendency (Figure 5B). These results imply that the decreased NF-κB in activated microglia with GBP2 suppression induces the reduced expression of pro-inflammatory cytokines. 

To confirm this change, we examined the production of pro-inflammatory molecules by the NF-κB-mediated signal pathway, which is a regulator of inflammatory cytokines. After transfection of the shD-GBP2 to the BV2 microglial cells, LPS was stimulated and various inflammatory cytokines at the transcriptomic level were measured by qRT-PCR using the mRNA extracted from the cells. As shown in Figure 6A–C, LPS induced a significant increase in IL-6, TNF-α, and iNOS in the BV2 microglia when compared to the control group, as is known. In contrast, the expression of pro-inflammatory molecules was remarkably decreased in cells treated with shD-GBP2. This tendency was also observed in the cells treated with BV2 microglial cells at the protein level (Figure 6D–F). IL-6, TNF-α, and iNOS expression in shD-GBP2-treated cells significantly induced lower M1 microglia-associated inflammatory factors than those in the cells treated with LPS or shScr. LPS dramatically induced the expression of pro-inflammatory cytokines. In addition, the shD-GBP2 pre-treatment of activated microglial cells upregulated the expression of IL-10, an inflammatory factor associated with M2 microglia (Figure 6G). Taken together, these results indicate that the *GBP2* gene plays an important role in regulating LPS-induced neuroinflammation and that the inhibition of the *GBP2* gene has an anti-inflammatory function in activated microglia.

### 3.6. Bioenergetic Profile in Mitochondria of Neuroblastoma by Activated Microglia

In previous experiments, we demonstrated that activated microglia induce an immune response leading to damage in neuronal cells. There are reports that the damage to nerve cells occurs due to changes in neuronal energy metabolism induced by cytokines released by M1 microglia [29,30]. As shown in Figure 6, we confirmed that *GBP2* induces a decrease in immune cytokines, and we hypothesized that this would reduce the induction of changes in neuronal energy metabolism, thereby preventing neuronal cell death. For this, a glycolysis stress test (GST) was performed on the Seahorse XF analyzer to assess changes in the glycolytic flux in neurons (Figure 7). This assay measures net glycolysis-dependent proton production in cells. It is anticipated that the reserve biochemical capacity of the enzyme for glycolysis is exceeded, resulting in the depletion of NAD^+^ and, ultimately, the inhibition of the entire pathway. The extracellular acidification rate (ECAR) was monitored in SH-SY5Y neuronal cells after co-culture with BV2 microglial cells. These data showed a clear concentration-dependent decrease in the glycolytic flux (Figure 7A), consistent with the anticipated inhibition of the proton efflux into the media. The glycolysis and glycolytic capacity were significantly increased following co-cultures of the neurons with BV2 microglia activated by LPS, but not in the NC group or co-culture neuron with shD-GBP2 treated microglia (Figure 7B). In addition, the capacity of the cell to switch from oxidative phosphorylation to glycolysis following the injection of oligomycin, a specific inhibitor of ATP synthase, is suppressed in co-cultured neurons with shD-GBP2-treated microglia, indicating that GBP2 Kd induces a more energetic profile in neurons stimulated by activated microglia. These results indicate that activated microglia lead to neuronal cytotoxicity through mitochondrial dysfunction in neuronal cells.

### 3.7. Increased Neuron Survival with GBP2 Knockdown in BV2 Microglial Cells

LPS-induced neuroinflammation was reduced by GBP2-shRNA in microglial cells. Based on previous results, we investigated whether downregulation of GBP2 could inhibit neuronal cell death. To address this, an MTT assay was conducted in a co-culture system using shSCR-or shD-GBP2 treated BV2 microglial cells after LPS treatment. Both the LPS and shScr treatments induced the increased cell death in SH-SY5Y neuronal cells, demonstrating that neurotoxicity occurred in activated microglia-induced neurotoxicity (Figure 8A). On the other hand, neuronal damage mediated by shD-GBP2 was decreased compared to that in other groups, indicating that the downregulation of GBP2 has a neuroprotective effect by suppressing the neurotoxicity associated with the excessive microglial pro-inflammatory response.

The induction of apoptosis is a major mechanism in cell death. Therefore, to assess which cell death mechanism was caused by neurotoxicity reduced by GBP2 Kd, we conducted a Wright–Giemsa staining experiment to visually observe cell death. As shown in Figure 8B, compared with the normal morphology of the control cells (NC group), the LPS-treated activated BV2 + Neuron group exhibited morphological characteristics of apoptosis such as sublobes, fragmented shapes, and apoptotic body formation. In comparison, the shD-GBP2-treated activated BV2 + Neuron group had a remarkably reduced number of morphological cells. 

To further quantify these results, we measured the rate of apoptosis using FACS analysis. As shown in Figure 8C,D, BV2 microglial cells significantly induced enhanced apoptosis (annexin V-positive) in SH-SY5Y neuronal cells upon LPS stimulation, but this effect was significantly suppressed by shD-GBP2-treated BV2 cells. The total cell death population, including apoptosis and necrosis, was higher in the LPS-treated cells (8.26%) than in the NC group (1.9%), whereas that of the shD-GBP2 group was 2.35% lower than that of the LPS group. These results demonstrate that the GBP2 Kd in the activated microglia stimulated by foreign stimuli elicits enhanced neuronal cell survival and can prevent neuronal cell damage against an over-activated immune response.

## 4. Discussion

In recent years, interest in the role of microglia in the brain has increased dramatically with regard to aging and neurodegenerative diseases such as Alzheimer’s [31] and Parkinson’s disease [32]. The main causes for these problems is the high susceptibility of neurons to inflammatory damage [33,34] Amyloid-beta or tau, known to be the causes of Alzheimer’s disease, a representative brain disorder, are being studied as target molecules for its treatment. Drugs targeting these proteins have been developed and are being used in clinical settings. However, since these drugs only slow the progression of the disease and are not fundamental cures, there is a need for more advanced treatments for Alzheimer’s disease. To achieve this, research is underway to develop carriers capable of delivering drugs directly to the brain by crossing the blood–brain barrier (BBB), and to discover new target proteins to enhance therapeutic efficacy. Recently, more fundamental therapeutic strategies are being pursued, such as controlling the activation of microglia [35,36]. The main role of microglia is as mediators of the inflammatory process [6]. However, the relationship between the brain and the immune system is complex and not fully understood [13]. In the injured brain, activated microglia cells participate in the course of inflammation, a process that includes the actions of various pro-inflammatory cytokines [36]. When microglia are stimulated with LPS, they can be activated, which promotes switching from a non-inflammatory and regulatory M2 phenotype to a pro-inflammatory M1 phenotype [37,38]. The M1 phenotype, which is the proinflammatory state of activated microglia, secretes various kinds of cytokines, such as IL-6 and TNFα, forming a vicious cycle of pro-inflammatory responses that continuously damage neurons and other important nervous system structures [39,40]. In addition to M1 polarization, macrophages undergo transcriptional rewiring and metabolic reprogramming [18]. For instance, M1 polarized macrophages activated with LPS display a metabolic shift from oxidative phosphorylation to aerobic glycolysis.

To better understand the cross-talk between the immune intruders and resident microglia and the contribution of the latter to neuroinflammation destruction and healing processes, we focused on gene expression when microglia are stimulated by foreign pathogens. Among the genes expressed in activated microglia, the *GBP2* gene is known to be a protein that is expressed and activated in the early phase of the inflammatory response and plays an important role in controlling immune responses [25,41]. It has also been used as a marker for IFN responsiveness in both cells and organisms because they are some of the most highly expressed genes after INF-γ stimuli and are induced within a few minutes of transcription induction by INF-γ. Therefore, we speculated that the *GBP2* gene may be a causative factor in inducing neuroinflammation in neurodegenerative diseases like Alzheimer’s disease and confirmed the expression of *GBP2* in Alzheimer’s disease animal models (Figure 1). These results suggest that *GBP2* has a high potential as a targeted biomarker for diagnosing and treating brain diseases in Alzheimer’s disease mouse brain tissues. Based on tissue data, we assumed that GBP2 is involved in the early immune response and may impact neuronal damage in the immune response of LPS-stimulated microglial cells. To address this, we conducted an experiment to activate microglial cells using LPS under various conditions (Figure 2). The results were consistent with a previous report that found that LPS activates microglia [42]. Furthermore, we confirmed that the expression of GBP2 is induced by these stimuli. 

When microglia encounter foreign pathogens in the brain, they choose whether to remove them or protect the brain [9,12], and then switch to the M1 phenotype to express pro-inflammatory cytokines if it is the former, or M2 type for anti-inflammation if it is the latter. LPS stimulates the pro-inflammatory reactions of microglia [18]. This fact was also confirmed by our results, leading to neuronal cytotoxicity due to the M1-type polarization of microglia by LPS stimulation (Figure 3); however, it is not clear which molecules are secreted from the activated microglia that affect neurons, and a more precise experiment is required to evaluate this. Overactivated microglia result in neuronal cell death in the brain as a side effect. To control the immune response of microglia, microglia were treated with shRNA, resulting in Kd of the *GBP2* gene. This study provides important information on the neuronal inflammatory response induced by activated BV2 microglia by knocking down the *GBP2* gene (Figure 4). When the *GBP2* gene is downregulated, activated microglia secrete less pro-inflammatory cytokines because of the switching from the M1 to M2-type polarization (Figure 4), indicating that this reduced the activation of the NF-kB pathway. The decreased activity of NF-κB by GBP2 Kd could cause the downregulation of pro-inflammatory cytokines. With regard to this, it also showed that the down-regulation of GBP2 potentially occurred via the NF-kB pathway, which can reduce NO and ROS production and alter phagocyte activity (Figure 5 and Figure 6), including a significant reduction in iNOS expression. These results suggest that GBP2 Kd action on activated microglia functions as an anti-inflammatory reaction.

During homeostasis, neurons actively induce anti-inflammatory regulation over microglia; however, when environmental or generic factors perturb these neuron-mediated anti-inflammatory signals, microglia can promote neuroinflammation, leading to neuronal toxicity. The neurotoxic effect induces a transition in neuronal metabolism from oxidative phosphorylation to glycolysis [43]. This metabolic shift is induced by substances secreted from M1 microglia. Therefore, we evaluated the changes induced in neuronal mitochondria by activated microglia. The glucose metabolism in neurons stimulated by activated microglia changed due to increased glycolysis and glycolytic capacity, leading to neuronal damage (Figure 7). On the other hand, Neurons stimulated by GBP2 Kd microglia exhibited decreased glycolytic function due to the inhibition of proton efflux. These changes were evaluated quantitatively using FACS analysis to assess their impact on neuronal cell survival (Figure 8). As expected, the data showed that GBP2 Kd was effective in reducing neuronal cell death in LPS-activated microglia. 

In summary, our study reveals the role of *GBP2* in the regulation of LPS-induced neuroinflammation in microglia and is the first report showing that the downregulation of GBP2 represents an anti-inflammatory function in microglia. Research on the regulation of neuroinflammation in neurodegenerative diseases has been relatively recent, thus requiring significant exploration into potential target molecules. From this perspective, our study demonstrates that the downregulation of *GBP2* can minimize neuronal damage caused by the induction of anti-inflammatory functions in activated microglia, providing evidence supporting *GBP2* as a potential candidate molecule for alleviating neurodegenerative diseases. However, for the clinical application of GBP2, further research is needed to determine whether regulating GBP2 gene expression in microglia shows positive effects in actual disease animal models. Nonetheless, this gene is of great importance as it provides a foundation for studying the cellular biological functions related to brain diseases and microglia. If molecular biological studies utilizing this gene and the actual expression of GBP2 in clinical patients are confirmed, this gene could become a highly valuable target for the prevention and diagnosis of brain diseases.

## Data Availability

Data are contained within the article and Appendix A.

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
