# Peer review of "Exploring the Role of Guanylate-Binding Protein-2 in Activated Microglia-Mediated Neuroinflammation and Neuronal Damage"

_biomedicines, 2024, doi:10.3390/biomedicines12051130_

Round 1

Reviewer 1 Report

Comments and Suggestions for Authors

GENERAL COMMENTS

The manuscript “Exploring the Role of Guanylate-Binding Protein-2 in the Acti-2 vated Microglia-Mediated Neuroinflammation and Neuronal 3 Damage” is an interesting and relevant scientific article since GBP2 plays a crucial role in the regulation of neuroinflammation in microglia by presenting an anti-inflammatory action. All of this leads to a reduction in neuronal damage and an increase in pro-inflammatory cytokines. GBP2 would be a potential candidate to alleviate neurodegenerative and inflammatory diseases.

SPECIFIC COMMENTS

1.     Aims should be included in the last paragraph of the introduction.

2.     In figure 6, why is there so much variability? explain it

3.     Has the microglia profile in M2 been analyzed such as CD163, TREM2, YM1..? Although the profiles of these cells are diverse, they express mediators or receptors with the ability to negatively regulate, repair or protect the body from inflammation and it would be interesting to analyze it in more detail.

4.     It would be interesting if the conclusions highlighted the importance of the study as well as its application in the clinic.

5.     Limited insights: While the authors did an excellent job summarizing the current findings, they should also provide valuable insights into the current status of the challenges faced in the field, and future directions.

Reviewer 2 Report

Comments and Suggestions for Authors

In this manuscript, You et al. attempt to explore the role of Guanylate-Binding Protein-2 (GBP2) in activated microglia during LPS-induced neuroinflammation. Although several genome-wide association studies indicate that the upregulation of the GBP2 gene is linked to Alzheimer's disease (Yang et al., 2015 (Curr Alzheimer Res); Cheng et al., 2014 (Biomed Res Int)), the role of GBP2 in microglia remains unknown.

The introduction section needs some improvement in terms of structure and flow. More emphasis should be placed on explaining why GBP2 gene might play a role in protecting neurons in the presence of neuroinflammation.

I would recommend the authors include the characterization of brain-region specific GBP2 gene expression levels in microglia in Figure 1. A comparison between control and AD mice would be helpful to explain why GBP2 could be a potential target for treating Alzheimer's disease. Which AD mouse did you use? It is unclear from the Materials and Methods section.

The authors mainly study the role of GBP2 at the genomic level. I don’t see a sufficient amount of work for understanding the phenotypic function of GBP2. The co-culture study in Figure 6 does not clearly show that transfecting shRNAs-GBP in microglia prevents the LPS-induced glycolytic flux in neurons. On the other hand, the authors do not directly show GBP2 affects the function of mitochondria in the microglia. Therefore, the authors should not argue that downregulation of GBP2 alters the function of mitochondria in the microglia.

In 'Scheme 1', the explanation of the key findings in this manuscript is not accurate. The authors do not assess the immune activation and use Tau-dependent neurodegeneration models in this study. Could you clarify what is meant by GBP2 down-regulation?

In my opinion, poor data presentation and a lack of method description make it difficult to provide a clear recommendation for the current version of the manuscript. The main issue is the lack of concrete evidence to support the manuscript's main conclusion.

Other comments:

1.      All genes should be written in italics.

2.      Abstract-“This study suggests that the GBP gene can be used as a therapeutic 29 target biomarker for inflammation-related to neurodegenerative diseases”. It should be GBP2 gene.

3.      Details of statistical analysis should be included in the figure legends. Also, I cannot find any information talking about how many replicates have been performed for each experiment.

4.      Potential limitations of the study should be clearly indicated in the discussion section.

Comments on the Quality of English Language

Overall, the paper's organization needs improvement. Minor editing of English is also required.

Round 2

Reviewer 1 Report

Comments and Suggestions for Authors

I consider that the authors have correctly addressed the comments and revisions and therefore the manuscript is suitable for publication.

Reviewer 2 Report

Comments and Suggestions for Authors

The authors have adequately addressed my comments.